# DATASETRESEARCH: BENCHMARKING AGENT SYSTEMS FOR DEMAND-DRIVEN DATASET DISCOVERY

## ABSTRACT

The rapid advancement of large language models has fundamentally shifted the bottleneck in AI development from computational power to data availability—with countless valuable datasets remaining hidden across specialized repositories, research appendices, and domain platforms. As reasoning capabilities and deep research methodologies continue to evolve, a critical question emerges: can AI agents transcend conventional search to systematically discover any dataset that meets specific user requirements, enabling truly autonomous demand-driven data curation? We introduce DATASETRESEARCH, the first comprehensive benchmark evaluating AI agents' ability to discover and synthesize datasets from 208 real-world demands across knowledge-intensive and reasoning-intensive tasks. Our tri-dimensional evaluation framework reveals a stark reality: even advanced deep research systems achieve only 22% score on our challenging `DatasetResearch-pro` subset, exposing the vast gap between current capabilities and perfect dataset discovery. Our analysis uncovers a fundamental dichotomy—search agents excel at knowledge tasks through retrieval breadth, while synthesis agents dominate reasoning challenges via structured generation—yet both catastrophically fail on "corner cases" outside existing distributions. These findings establish the first rigorous baseline for dataset discovery agents and illuminate the path toward AI systems capable of finding any dataset in the digital universe. Our benchmark and comprehensive analysis provide the foundation for the next generation of self-improving AI systems. The code and dataset will be open-sourced soon.

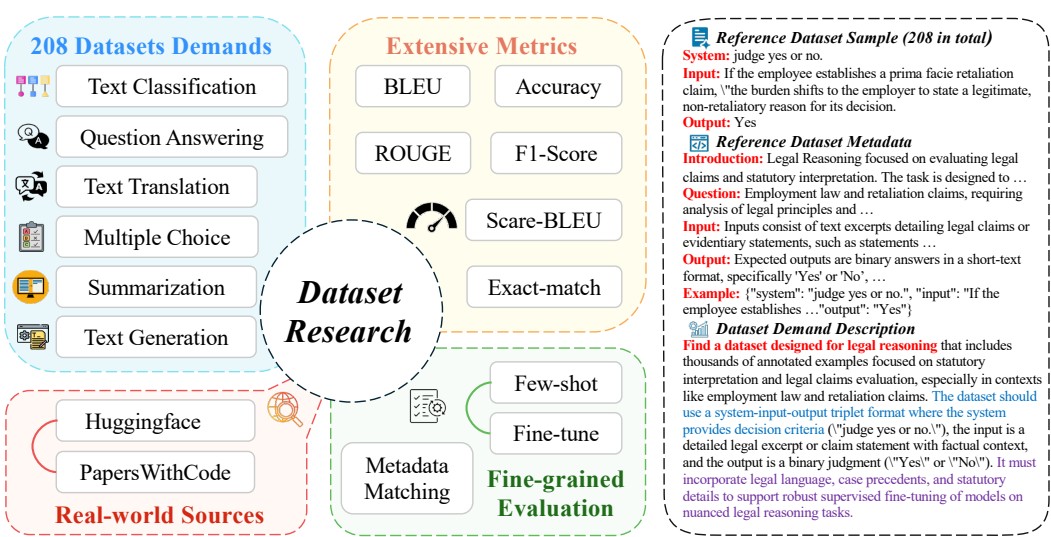

Figure 1: **Overview of DATASETRESEARCH:** A Benchmark for Dataset Discovery Agents. It features two sources, 208 dataset demands, three evaluation methodologies, and six NLP metrics.

# 1 INTRODUCTION

The advancement of artificial intelligence (AI) (xAI, 2025a; OpenAI, 2025b; Anthropic, 2025; Comanici et al., 2025; Guo et al., 2025) has increasingly positioned datasets as foundational assets for scientific discovery and technological progress. Contemporary powerful reinforcement learning-based agentic systems exhibit a strong dependence on high-quality datasets for optimal performance (Ye et al., 2025; Wang et al., 2025; Sun et al., 2024; Wang et al., 2024; Team et al., 2025; Mai et al., 2025; Yang et al., 2025b; Xie et al., 2025; Wei et al., 2025). The identification and synthesis of appropriate datasets represents a critical bottleneck in the initiation of scientific research endeavors. The traditional research workflow—problem identification, dataset requirement formulation, manual data search, synthesis, and model training—represents an increasingly antiquated approach in an era of rapidly evolving AI capabilities. As reasoning methodologies and deep research tools continue to advance, a fundamental question emerges: can AI agents transcend the limitations of conventional search to systematically discover any dataset that meets specific user requirements from this vast, largely untapped data universe?

While previous work has explored demand-driven dataset discovery and synthesis (Viswanathan et al., 2023; Walker et al., 2023; Gandhi et al., 2024), these approaches represent initial steps that do not fully harness the deep reasoning and inferential capabilities enabled by modern large language models. The emergence of sophisticated reasoning agents and deep research methodologies demands a more rigorous evaluation of whether current systems can achieve the ambitious goal of comprehensive, demand-driven dataset discovery across the entire digital landscape. To address this critical gap, we introduce DATASETRESEARCH, the first comprehensive benchmark designed to systematically evaluate whether AI agents can make any relevant dataset discoverable based on specific user demands. As the inaugural benchmark in this domain, DATASETRESEARCH exhibits several distinctive characteristics shown in Figure 1:

- **Comprehensive Coverage and Rich Examples:** We curated 208 data requirements across six major NLP tasks. Using OpenAI o3 (OpenAI, 2025b), we generated comprehensive metadata for each, classifying them as knowledge- or reasoning-based (Table 1), and created corresponding query pairs to serve as agent inputs.
- **Thorough Evaluation with Diverse Baselines:** Our integrated evaluation framework assesses data quality from baselines including search, synthesis, and deep research agents. Evaluation methodology combines three approaches: metadata similarity, few-shot learning, and supervised fine-tuning with LLaMA-3.1-8B (Dubey et al., 2024), yielding a normalized performance score across all tasks.
- **Stratified Difficulty with Reference Subsets:** To manage high computational costs, we created `DatasetResearch-pro`, a challenging subset of 20 tasks from DATASETRESEARCH. These tasks were selected as the most difficult for a baseline agent (GPT-4o-search-preview) to effectively differentiate the capabilities of advanced systems.

Comprehensive experiments on DATASETRESEARCH demonstrate that current agent systems fall considerably short of optimal performance, with even the most advanced deep research systems achieving a maximum score of merely 22% on our evaluation subsets. Our analysis further reveals a pronounced performance differentiation pattern: search agents leverage their robust information retrieval capabilities to excel in knowledge-based tasks, while synthesis agents capitalize on their capacity for constructing reasoning pathways to demonstrate superior performance in reasoning-based challenges. These findings not only expose the limitations of existing technologies but also highlight the tremendous potential of automated data synthesis approaches.

To conclude, we make the following contributions: (1) We present DATASETRESEARCH, the first comprehensive benchmark for demand-driven dataset discovery and synthesis, featuring 208 real-world requirements. Our multi-dimensional evaluation methodology assesses metadata alignment, few-shot performance, and supervised fine-tuning effectiveness across six task categories. (2) Extensive experiments on state-of-the-art systems reveal significant limitations, with top scores of only 0.2 on our challenging `DatasetResearch-pro` subset. We identify a clear specialization: search agents excel at knowledge-based tasks, while synthesis agents are superior for reasoning-based challenges. (3) We provide the first systematic analysis of failure modes in automated dataset construction, showing that all current methods struggle with out-of-distribution corner cases, which highlights fundamental challenges in generalization.

## 2 RELATED WORK

### 2.1 DATASET DISCOVERY AND SYNTHESIS

The critical role of high-quality datasets is well-recognized in recent research (Ye et al., 2025; Wang et al., 2025; Yang et al., 2025b; Xie et al., 2025; Chen et al., 2024), leading to two primary acquisition strategies: discovery and synthesis. Dataset discovery methods include bi-encoder retrievers (Viswanathan et al., 2023; Soylu et al., 2024) and conversational AI (Walker et al., 2023; Majumder et al., 2024; Gu et al., 2024; Yang et al., 2025c), though the latter can hallucinate datasets. Dataset synthesis involves transforming existing data (Gandhi et al., 2024) or using agentic methods for construction, which has achieved strong results on benchmarks like SWE-bench verified (Jimenez et al., 2023; Yang et al., 2025b; 2024; Hui et al., 2024). The emergence of agents capable of iterative reasoning and search (Zheng et al., 2025; Singh et al., 2025) necessitates a new standard for evaluation, motivating our development of DATASETRESEARCH.

### 2.2 AGENTS WITH SEARCH AND REASONING CAPABILITIES

AI agent evolution is driven by both stronger foundation models (xAI, 2025a; OpenAI, 2025b; Anthropic, 2025; Comanici et al., 2025; Guo et al., 2025; Yang et al., 2025a) and sophisticated search and reasoning abilities (Zheng et al., 2025; Jin et al., 2025; Song et al., 2025; Lu et al., 2024). DATASETRESEARCH evaluates both types of agents. We assess search-enabled agents, including systems with integrated search tools (OpenAI, 2025a) and deep research systems (OpenAI, 2025c; Google, 2025; xAI, 2025b). For reasoning-based synthesis, we evaluate leading agents like OpenAI o3 (OpenAI, 2025b), Gemini 2.5 Pro (Comanici et al., 2025), Claude 4 (Anthropic, 2025), QwQ-32B(Team, 2025), and Grok 4 (xAI, 2025a). In particular, we use OpenAI o3's (OpenAI, 2025b) reasoning to construct challenging dataset generation tasks.

## 3 DATASETRESEARCH

In this section, we systematically detail the construction of the DATASETRESEARCH benchmark, including its composition and the comprehensive evaluation protocol designed to assess agentic systems. DATASETRESEARCH curation pipeline is shown in Figure 2, and evaluation pipeline is shown in Figure 3.

### 3.1 TASK DEFINITION

In AI developing workflows, practitioners frequently encounter the challenge of identifying and collecting appropriate datasets that align with specific training requirements. This process, which we term **data discovery**, involves both systematically searching through available data resources and synthesizing new datasets as needed to satisfy given criteria and constraints. Formally, we define the data discovery task as follows: Given a natural language **demand description** $D$ that specifies the desired characteristics of a dataset, a **DataResearcher** for data discovery must output a **discovered dataset** $S_d = \{d_1, d_2, ..., d_n\}$ that optimally satisfies the specified demand $D$.

To evaluate this task, we establish a benchmark framework based on **MetaTriplets**, where each MetaTriplet $M_i = (D_i, S_{r_i}, \text{Meta}_{r_i})$ consists of three components: (1) a **demand description** $D_i$ representing a real-world data collection requirement expressed in natural language, (2) a **reference set** $S_{r_i} = \{d_{r_1}, d_{r_2}, ..., d_{r_k}\}$ containing ground truth datasets that satisfy the demand $D_i$, and (3) a **reference metadata** $\text{Meta}_{r_i}$ providing detailed information about each dataset in $S_{r_i}$, including domain specifications, format descriptions, quality metrics, and other relevant characteristics.

The evaluation process compares the data researcher's discovered dataset $S_d$ against the reference dataset $S_r$ using evaluations that assess both metadata relevance by generating a **Discovered Metadata** and downstream task performance by testing discovered dataset $S_d$ on the reference dataset $S_r$. This triplet-based evaluation framework enables systematic assessment of DataResearcher across diverse domains and requirements, facilitating the development of AI-driven solutions that can autonomously identify and provision datasets for AI model training—thereby establishing a self-improving ecosystem where artificial intelligence systems enhance their own data discovery and curation capabilities.

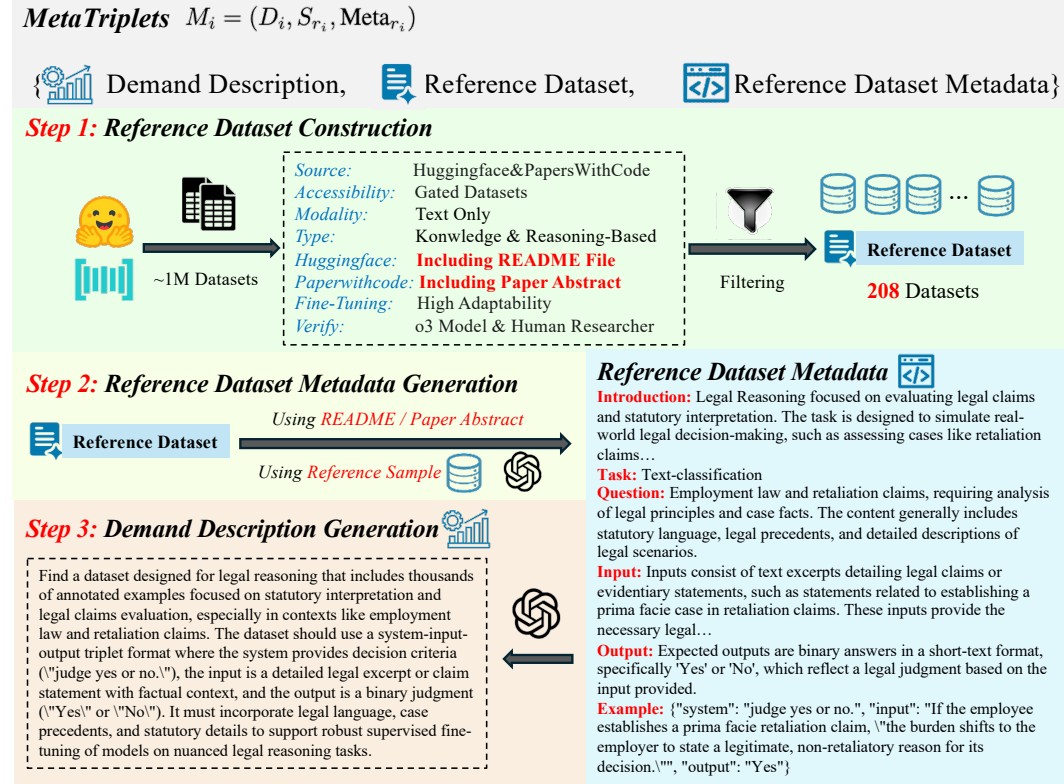

Figure 2: **Curation pipeline of the DATASETRESEARCH benchmark.** From an initial dataset of over 1 million candidates, we first apply a series of filtering rules to curate a final reference set of 208 instances. We then utilize the state-of-the-art o3 model to process the associated README files and data samples, generating metadata across six distinct dimensions. Finally, the o3 model synthesizes this metadata to generate the corresponding dataset demands.

## 3.2 DATASETRESEARCH CURATION

DATASETRESEARCH collects requirements corresponding to 208 real-world datasets shown in Table 1, with 91 sourced from HuggingFace and 117 from Papers with Code. Each dataset demand in DATASETRESEARCH is accompanied by three key components: a detailed **demand description**, the corresponding **reference dataset**, and comprehensive **reference metadata** associated with the reference dataset.

**Data Collection Pipeline** We develop a systematic collection methodology guided by three key principles: ensuring real-world authenticity of dataset demands, maintaining automated evaluation feasibility, and preserving structural clarity for agent processing. Our collection process, shown in Figure 2 and exemplified through HuggingFace datasets, follows a multi-stage filtering and refinement approach designed to produce high-quality, evaluable demands.

- **Step 1: Initial Curation** We use the HuggingFace API to identify all "gated" datasets, which require manual approval for access. We select these as our starting point to mitigate data leakage, as search agents cannot automatically download and process these datasets even if they are identified.
- **Step 2: Task and Modality Filtering** We filter this collection to retain only text-modality datasets whose annotated task fell within one of six categories: question-answering, text-summarization, text-classification, text-generation, multiple-choice, or language-translation. This step ensures the feasibility of an automated evaluation pipeline. We further exclude tasks where baseline model performance had already reached near-saturation levels, as these provide insufficient discriminative capacity for meaningful evaluation. This comprehensive filtering process yields 422 datasets.
- **Step 3: Documentation Quality Check** We further filter for datasets that contain comprehensive and informative README files, which serve as crucial references for generating reference metadata and demand descriptions. For demands from Papers with Code, we utilize the abstracts of

Table 1: DATASETRESEARCH comprises 208 dataset demands derived from six categories of real-world NLP datasets from two distinct sources, evaluated using diverse metrics. The reference datasets in DATASETRESEARCH are divided into knowledge-based and reasoning-based tasks. `DatasetResearch-pro` is a subset containing 20 more challenging examples.

| Task | Metric | Num(Knowledge) | | Num(Reasoning) | | Num(pro) |
|------|--------|:----:|:----:|:----:|:----:|:----:|
| | | 🙌 | [▥] | 🙌 | [▥] | |
| Multiple Choice | Accuracy | 9 | 4 | 4 | 10 | 5 |
| Text Generation | BLEU | 6 | 2 | 9 | 21 | 3 |
| Text Summarization | ROUGH | 2 | 1 | 0 | 8 | 3 |
| Question Answering | F1, Exact Match | 4 | 9 | 10 | 25 | 3 |
| Text Classification | Accuracy | 9 | 3 | 23 | 23 | 3 |
| Language Translation | BLEU | 1 | 1 | 14 | 10 | 3 |

corresponding papers and dataset samples instead of README files. This step results in 261 candidates.
- **Step 4: Fine-Tuning Suitability** We select datasets amenable to fine-tuning, excluding those designed purely for pre-training or lacking clear label columns. This leaves 104 suitable datasets.
- **Step 5: Automated Reformatting** For each of the 104 datasets, we prompt the OpenAI o3 model to propose a fine-tuning format template by analyzing the README and data samples (e.g., combining specific columns into input and output fields, adding a task-specific instruction).
- **Step 6: Human Verification** We manually review and refine these suggestions, adding instructions where necessary and removing a few datasets deemed unsuitable for fine-tuning. This process finalizes a set of 91 high-quality datasets from HuggingFace.
- **Step 7: Demand Description and Metadata Generation** For each of the 91 datasets, we use the o3 model to generate a comprehensive metadata profile, including an introduction, domain, input schema, output schema, and sample count. Based on this metadata, we then prompt OpenAI o3 to generate natural language demand descriptions that serve as inputs for our **DataResearcher**.

These 91 demands, combined with the 117 generated via a similar process from Papers with Code, form the 208 tasks in DATASETRESEARCH. From this pool, we curate a specialized subset of 20 particularly challenging tasks to create `DatasetResearch-pro`. This subset is constructed to probe the limits of current agents by selecting the 20 tasks where GPT-4o-search-preview achieved the lowest scores in the fine-tuning setting. On this highly difficult subset, we expand our evaluation to include the most advanced deep research agents. Detailed prompts used for metadata and demand generation are available in the Appendix.

**Categorizing Knowledge-Based and Reasoning-Based Tasks** To distinguish between tasks where reference dataset needs to be more oriented toward factual knowledge learning versus those requiring reasoning-based logical learning, we categorize these requirements into two types. Based on dataset characteristics and corresponding specific requirement descriptions, we manually identify and annotated 51 knowledge-based tasks and 157 reasoning-based tasks. **Knowledge-based** tasks require data coverage of extensive factual information, structured knowledge, and predefined classification systems, emphasizing the breadth and accuracy of the constructed data. In contrast, **reasoning-based** tasks require data that can guide the construction of reasoning pathways and logical relationships from input to output, emphasizing the achievement of cross-domain problem-solving capabilities and cognitive generalization through learning reasoning patterns rather than relying on precise coverage of domain knowledge in reference dataset.

### 3.3 EVALUATION METHODOLOGY

We employ a comprehensive evaluation methodology shown in Figure 3 that assesses dataset quality from both intrinsic metadata characteristics and extrinsic performance on downstream tasks.

**Metadata-Based Evaluation** To assess the primary data quality of discovered datasets, we score the semantic alignment between reference metadata and discovered metadata. Using OpenAI o3 as a judge (Zheng et al., 2023), we assign a score from 0 to 10 for each metadata dimension, including introduction, task, question, input, output, and example. Detail description is shown in Section D. The final metadata score is the average of these dimensional scores. Critically, because the OpenAI o3 is

Figure 3: **DATASETRESEARCH evaluation methodology.** We evaluate data quality using LLaMA-3.1-8B model on three metrics: metadata similarity (via o3), plus fine-tuning and few-shot performance. The latter two are measured by the performance ratio of discovered to reference data $(S_{ft}/S_{ref})$, normalized by a zero-shot baseline $(S_{base}/S_{ref})$.

used to generate both the reference and the discovered metadata, using it for evaluation systematically mitigates potential scoring biases.

**Downstream Task Performance Evaluation** In order to assess the practical gains of discovered datasets on real-world tasks, we assess the practical utility of the datasets via test-time and training-based methods. For the six task categories in DATASETRESEARCH, we design and employ six corresponding metrics. We evaluate performance across three settings:

- **Zero-shot Baseline:** We directly evaluate LLaMA-3.1-8B on the reference set without any fine-tuning or in-context examples to establish a performance floor.
- **Few-shot Learning:** We provide 1, 3, and 5 examples from the discovered or synthesized datasets as in-context examples (Parnami & Lee, 2022) for LLaMA-3.1-8B and evaluate on full reference dataset.
- **Fine-tuning:** We fine-tune LLaMA-3.1-8B on the discovered or synthesized datasets with fixed hyperparameters and then evaluate its zero-shot performance on the reference set.

To ensure fair comparisons across tasks with different evaluation metrics, we implement a standardized normalization procedure for all performance scores. Given the heterogeneous nature of evaluation metrics across our six task categories—ranging from BLEU scores for translation tasks to accuracy for classification—direct comparison would be misleading without proper normalization.

The performance $S_{\text{ref}}$ of LLaMA-3.1-8B model fine-tuned directly on the reference set serves as the upper bound for score normalization, representing the theoretical maximum performance achievable with perfect data for each specific task. For each evaluation setting (few-shot or fine-tuned), the score $S_{\text{eval}}$ of a discovered dataset is normalized against this upper bound. The final normalized score is calculated as:

$$\text{Normalized Score} = S_{\text{eval}}/S_{\text{ref}}$$

This formula positions the agent's performance on a scale from 0 to 1, or higher if the discovered dataset is superior to the reference set, enabling a fair and direct comparison of agent capabilities across all tasks in DATASETRESEARCH.

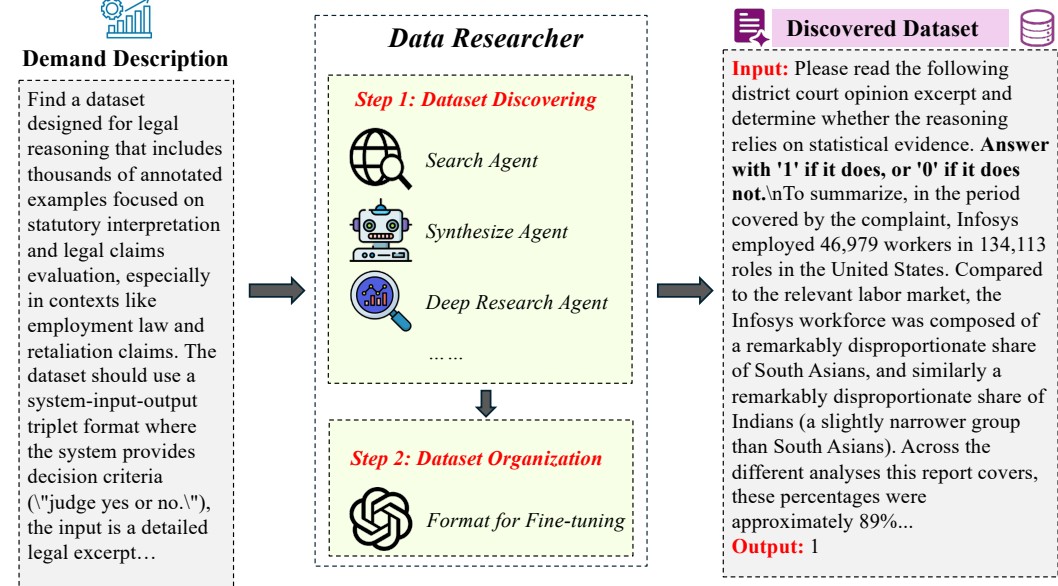

Figure 4: **Overview of the DataResearcher Baseline Workflow.** The workflow starts with anonymized dataset requirements. The DataResearcher module then discovers or synthesizes a matching dataset. Subsequently, OpenAI o3 generates metadata for this dataset, which is then compared against the reference metadata, as illustrated in Figure 3.

## 4 DATARESEARCHER

To evaluate AI systems' capabilities in generating data for AI training, we construct three distinct baseline Data Researchers shown in Figure 4 for DATASETRESEARCH: search agents, synthesis agents and deep research agents. These agents represent different paradigmatic approaches to data construction and are implemented as follows:

- **Search Agents** take a natural language demand, query the HuggingFace Hub for the top five public datasets, and programmatically select the first valid one. Our benchmark's gated datasets prevent finding the original reference.
- **Synthesis Agents** use OpenAI o3 to generate 500 data samples based on a demand description. This is performed in two settings: with one reference sample provided for guidance (w/ ref) and without any reference sample (w/o ref).
- **Deep Research Agents** employ deep research tools from OpenAI, Grok, and Gemini to perform a web-wide analysis and identify the best-fitting dataset. Due to a lack of API access, this process requires human-in-the-loop execution.

After obtaining the preliminary discovered dataset, we leverage OpenAI o3 to automatically parse and convert all samples into a standardized fine-tuning format with complete input and output pairs, ensuring compatibility with downstream training procedures and yielding the final discovered dataset ready for evaluation.

To ensure a fair comparison, the Data Researchers have the following experimental settings:

- Search agents evaluated on DATASETRESEARCH are restricted to public datasets on HuggingFace to facilitate automated data formatting and processing. For `DatasetResearch-pro`, search scope is expanded to the entire web, with results manually curated for relevance and accessibility.
- For all datasets returned by search agents, we use the OpenAI o3 to parse the data into a standardized fine-tuning format and to generate their corresponding metadata, mirroring the format of our reference set. If a discovered dataset contained more than 1000 samples, we truncate it to the first 1000 sample pairs.
- For synthesis agents, we generate datasets of 500 samples by prompting the model to produce 10 samples at a time and concatenating the results over 50 iterations. This mitigates potential quality degradation from long context windows.

Table 2: **Main results on the DATASETRESEARCH benchmark.** We report normalized few-shot and fine-tuning scores on Knowledge and Reasoning tasks using DTP (Downstream Task Evaluation), alongside metadata similarity scores. Synthesis-based agents (OpenAI o3) demonstrate superior performance on reasoning tasks, whereas search-based agents (GPT-4o-search) excel at knowledge-based tasks. Display Best results are displayed in **bold**, the second-best results with an underline.

| Agent | Method | DTP Evaluation | | Metadata Evaluation | | | | | | |
|---|---|---|---|---|---|---|---|---|---|---|
| | | *Knowledge(%)↑* | *Reasoning(%)↑* | Intro | Task | Ques | Input | Output | Example | Avg ↑ |
| *Baseline* | | | | | | | | | | |
| | | 10.41 | 11.84 | | | | | | | |
| *Search-based* | | | | | | | | | | |
| GPT-4o-search | 1 Shot | 9.82 | 7.25 | 5.7600 | 6.6300 | 5.3600 | 5.7700 | 5.3200 | 5.4100 | 5.7083 |
| | 3 Shots | 9.84 | 8.43 | | | | | | | |
| | 5 Shots | 10.22 | 8.70 | | | | | | | |
| | Fine Tune | **41.89** | 27.54 | | | | | | | |
| GPT-4o-mini-search | 1 Shot | 6.48 | 5.73 | 5.5000 | 6.4400 | 5.2900 | 5.7000 | 4.8800 | 5.3300 | 5.5233 |
| | 3 Shots | 6.45 | 7.89 | | | | | | | |
| | 5 Shots | 10.38 | 8.39 | | | | | | | |
| | Fine Tune | 12.12 | 17.35 | | | | | | | |
| *Synthesis-based* | | | | | | | | | | |
| OpenAI o3 w/ ref | 1 Shot | 10.25 | 17.38 | 8.6300 | 8.7100 | 8.1100 | 9.0100 | 9.3600 | 8.3200 | **8.6900** |
| | 3 Shots | 21.81 | 32.14 | | | | | | | |
| | 5 Shots | 23.91 | 28.92 | | | | | | | |
| | Fine Tune | 38.98 | **72.70** | | | | | | | |
| OpenAI o3 w/o ref | 1 Shot | 10.16 | 12.26 | 8.5800 | 8.7400 | 8.1100 | 8.8000 | 9.1700 | 8.0400 | 8.5730 |
| | 3 Shots | 17.25 | 25.53 | | | | | | | |
| | 5 Shots | 14.81 | 19.44 | | | | | | | |
| | Fine Tune | 37.94 | 67.25 | | | | | | | |

## 5 EXPERIMENTS

Our experiments are designed to rigorously assess the capabilities of DataResearcher in demand-driven dataset discovery. We present results on both the comprehensive DATASETRESEARCH benchmark and its challenging `DatasetResearch-pro` subset, revealing critical insights into the strengths and weaknesses of current search-based, synthesis-based and deep research approaches.

### 5.1 EXPERIMENTAL SETUP

For DATASETRESEARCH, we conduct comprehensive evaluations on search-based APIs including gpt-4o-search-preview and gpt-4o-mini-search-preview (OpenAI, 2025a), which provide real-time web search capabilities, as well as the synthesis capabilities of the advanced reasoning-based model OpenAI o3, known for its sophisticated analytical and generation abilities. For the more challenging `DatasetResearch-pro` subset, we assess the performance of state-of-the-art closed-source deep search agents that represent the current frontier in AI-powered research capabilities, including OpenAI Deep Research (OpenAI, 2025c), Gemini Deep Research (Google, 2025), and Grok Deep Research (xAI, 2025b).

### 5.2 MAIN RESULTS AND ANALYSIS

**Performance on the DATASETRESEARCH Benchmark** As shown in Table 2, agent performance on DATASETRESEARCH highlights a clear dichotomy in capabilities based on task-cognitive demands. For knowledge-based demands, search-based DataResearcher demonstrate significant advantages where GPT-4o-search agent achieves the highest fine-tuning score of 42%. Conversely, for reasoning-based tasks synthesis-based agents are undoubtedly superior, with the OpenAI o3 w/ ref agent attaining the highest fine-tuning score of 73%. Notably, we observe that few-shot evaluation results exhibit outcomes that are closely aligned with fine-tuning experiments across both task categories, which suggests a practical implication: since fine-tuning experiments are computationally expensive and time-consuming, few-shot evaluation can serve as an efficient preliminary assessment to rapidly detect and compare DataResearcher capabilities before committing to full fine-tuning procedures.

In metadata evaluation, we reveal that synthesis-based methods significantly outperform across output metrics, which reveals the core advantage of synthesis-based methods in fine-tuning tasks: their ability to generate more aligned output data, thereby providing models with sufficient learning material to master reasoning pathways from input to output. Furthermore, we identify that the primary factor limiting search-based methods' performance lies in the fact that retrieved existing datasets often

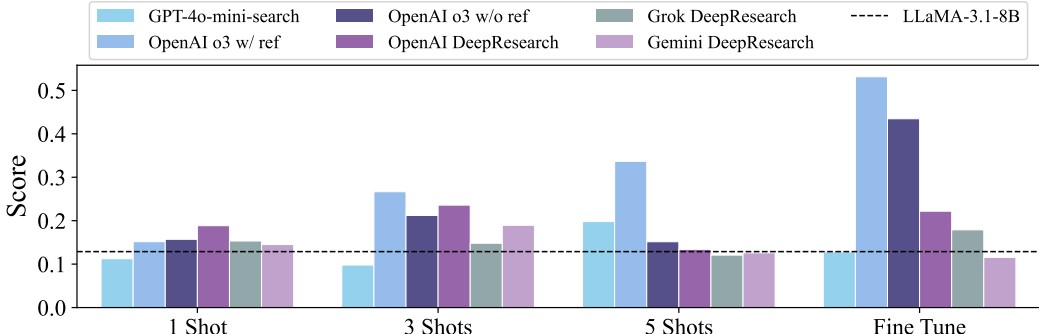

Figure 5: **Performance comparison of various agent systems on the `DatasetResearch-pro` subset across few-shot and fine-tuning settings.** Synthesis agents based on o3 perform exceptionally well, and the DeepResearch system generally outperforms the GPT-4o search system. The dashed line indicates the performance of LLaMA-3.1-8B baseline on test set.

cannot align with the instruction as precisely as synthesized data, resulting in relatively weaker data research instruction-following capabilities.

**Analysis of Learning Paradigms**  Our evaluation also conducts detailed analysis of different learning strategies. We demonstrate that 1, 3, 5-shot results maintain consistent relative trends with fine-tuning, although some 5-shot experiments fail to follow expected shot-scaling patterns. This is primarily attributed to the small-scale models we employ being unable to maintain effective long-range attention mechanisms when processing longer context windows (Schaeffer et al., 2023). Notably, among few-shot settings, 3-shot demonstrates the most stable and representative performance, achieving a favorable balance between computational efficiency and performance outcomes.

**Performance on the `DatasetResearch-pro` Subset**  To explore the performance boundaries of state-of-the-art deep research agents, we conduct experiments on the `DatasetResearch-pro` subset. Results shown in Figure 5 and Figure 9 demonstrate that advanced deep research systems such as OpenAI DeepResearch, achieving a score of 0.2218, significantly outperform standard search methods like GPT-4o-mini-search, yet their overall performance remains relatively modest. Meanwhile, we observe that synthesis-based methods also exhibit significant performance degradation on `DatasetResearch-pro`, achieving only scores approaching 0.5, indicating that tasks challenging for search-based approaches similarly pose substantial difficulties for synthesis methods.

**Analysis**  Our analysis reveals a key trade-off: search-based methods excel at sourcing diverse, knowledge-rich data for factual tasks, whereas synthesis-based methods are superior for generating logically coherent datasets for reasoning-intensive tasks. The iterative deep research methodology surpasses these single-shot approaches by discovering higher-quality and more comprehensive data, leading to significant performance gains. However, we identify a critical limitation where all current methods fail on niche "corner cases," as their performance is fundamentally constrained by the coverage of existing data distributions. The relevant case studies can be found in the appendix.

## 6 DISCUSSION

In this work, we introduced DATASETRESEARCH, a comprehensive benchmark to evaluate agentic systems on demand-driven dataset discovery. Our findings reveal a critical performance gap and a clear specialization: search-based agents excel at knowledge-intensive tasks, while synthesis-based agents dominate reasoning challenges, motivating several key future directions. A natural evolution is the development of hybrid agents that intelligently integrate search and synthesis to balance data quality and coverage. To achieve a true automated research workflow, these systems must also advance beyond structured repositories to automatically curate data from the unstructured web. Furthermore, to democratize these capabilities, a crucial next step is to explore powerful open-source LLMs for data synthesis, reducing the reliance on costly closed-source APIs and paving the way for the next generation of accessible, robust AI-powered research assistants.

ETHICS STATEMENT

This work adheres to the ICLR Code of Ethics. Our study does not involve human subjects, sensitive personal data, or experiments that could directly cause harm to individuals or communities. We have taken care to consider issues of fairness, privacy, and security when designing our methods and presenting our results. We are not aware of any potential conflicts of interest, legal compliance issues, or research integrity concerns related to this submission.

REPRODUCIBILITY STATEMENT

We have made every effort to ensure the reproducibility of our results. Details of the model architecture, training procedures, and evaluation protocols are provided in the main text and appendix. Hyperparameters, dataset preprocessing steps, and implementation details are described in the supplementary materials. To further support reproducibility, we upload the source code as supplementary material. These resources should allow other researchers to replicate our findings and build upon our work.

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

## A   THE USE OF LLMs

In the article, we only used LLMs to polish our writing, and did not use them for any other assistance.

## B   EVALUATION SETUP

We evaluate leading agents on our DATASETRESEARCH with 208 tasks and `DatasetResearch-pro` with 20 tasks benchmarks. The agents include search-based DataResearcher like GPT-4o-search, synthesis-based DataResearcher like OpenAI o3 with and without reference examples and deep research based DataResearcher like OpenAI DeepResearch. Evaluation is twofold: (1) Metadata-based Evaluation that reflects dataset discovery instruction-following capabilities, and (2) Downstream Task Performance (DTP) that reflects the overall data performance of the discovered set. In our experiments, we calculate six distinct metrics tailored to different task categories:

- **Accuracy** measures the proportion of correct predictions, reflecting models' ability to identify the correct option or label.
- **F1-Score** (Joshi et al., 2017) computes the harmonic mean of precision and recall, providing a balanced assessment of model performance, especially when dealing with partial matches or token-level evaluation.
- **Exact Match** (Joshi et al., 2017) evaluates the percentage of predictions that exactly match the ground-truth answers, offering a strict assessment criterion.
- **BLEU** (Papineni et al., 2002) measures n-gram overlap between generated and reference text, assessing the quality and fluency of generated content.
- **SacreBLEU** (Keenan, 2017) provides a standardized and reproducible version of BLEU scoring, ensuring consistent evaluation.
- **ROUGE** (Lin, 2004) calculates recall-oriented overlap of n-grams and longest common subsequences, specifically designed for evaluating text summarization quality and content preservation.

## C   ANALYSIS

This section delves into specific cases to provide a qualitative understanding of the performance patterns observed in our experiments. By examining individual examples from Figure 6, Figure 7 and Figure 8, we highlight the distinct behaviors of search, synthesis, and deep research methodologies, particularly focusing on the fine-tuning performance.

### C.1   KNOWLEDGE VS. REASONING

Figure 6 offers a clear contrast between how different methods handle knowledge and reasoning-based tasks (Ravichander et al., 2019). For reasoning-centric requirements, synthesis-based methods construct richly detailed data with explicit thought processes, which guides the fine-tuned model toward more logical analysis. In contrast, for knowledge-based tasks, search methods excel by retrieving diversified information with broader coverage and higher knowledge density. This enables the fine-tuned model to access richer knowledge sources, thereby exhibiting stronger knowledge coverage and response accuracy on the reference set.

This reveals a key behavioral pattern: the breadth advantage of search methods allows them to capture long-tail knowledge distributions, making models more robust for diverse, fine-grained factual queries. Conversely, the depth advantage of synthesis methods lies in the ability of powerful LLMs to construct highly structured, logically coherent reasoning data, making it ideal for training models on tasks that depend on generalizable logic rather than pure fact retrieval.

### C.2   DEEP RESEARCH METHODOLOGY

As observed in Figure 7, the deep research methodology, with its iterative information gathering and reasoning-guided exploration, retrieves data of significantly higher quality than single-shot search or generation. Under identical query conditions(Dao et al., 2023), the resulting dataset

*Reasoning: synthesis better!*

**Reference Set Sample (Accuracy: 0.037)**
**Huggingface dataset_id:**
**nguha/legalbench**
**Input:** When a defendant presents an affidavit concerning a factual narrative of the events that is neither contradicted by the record nor inherently incredible and the government offers nothing more than contrary representations to contradict it, the defendant is entitled to an evidentiary hearing.
**Output:** Yes

**Dataset Demand:**
Find a dataset designed for legal reasoning that includes thousands of annotated examples focused on statutory interpretation and legal claims evaluation, especially in contexts like employment law and retaliation claims. The dataset should use a system-input-output triplet format where the system provides decision criteria (\"judge yes or no.\"), the input is a detailed legal excerpt or claim statement with factual context, and the output is a binary judgment (\"Yes\" or \"No\"). It must incorporate legal language, case precedents, and statutory details to support robust supervised fine-tuning of models on nuanced legal reasoning tasks.

**OpenAI o3 Synthesis Dataset Sample (Accuracy: 0.4907)**
**Input:** System: Judge yes or no.\n\nClaim: An employee invokes the False Claims Act, stating he was fired for threatening to report fraudulent Medicare billing. Termination occurred six months after his threat, following documented performance issues: three written warnings, two for tardiness and one for failure to follow sterile-procedure protocols. Employer policy dictates termination after three infractions within a calendar year. Precedent (Greene v. Potter, 557 F.3d 765) holds temporal proximity weakens as months pass, and employers may prevail with clear, consistently applied disciplinary frameworks. Do these facts justify a conclusion that the firing was not retaliatory?
**Output:** No

**OpenAI gpt-4o Search Dataset Sample (Accuracy: 0.0)**
**Huggingface dataset_id:**
**mteb/LegalReasoningCausalityLegalBenchClassification**
**Input:** "Please read the following district court opinion excerpt and determine whether the reasoning relies on statistical evidence. Answer with \'1\' if it does, or \'0\' if it does not.\nIn her Opposition Brief, Carman points to statistics of the racial composition of Defendants' employees as evidence of discrimination (see Doc. 66 at 8, 29) (alleging that while Erie County is \"made up of approximately 12.2% African American individuals,\" Defendants only employ 4.5% Africans Americans in their \"top paid positions\"). However, this statistic is not relevant in evaluating Carman's discrimination claims. For statistics to serve as evidence of discriminatory hiring practices, the Sixth Circuit requires a comparison of \"the racial composition of the pool of qualified [assistant prosecutor / DJFS director] candidates in the [Erie County] market.\" Hopkins v. Canton City Bd. of Educ., 477 F. App'x 349, 358 (6th Cir. 2012). Carman's comparison falls short, failing to meaningfully identify the racial composition of the pool of qualified candidates."
**Output:** 1

*Knowledge: search better!*

**Reference Set Sample (Accuracy: 0.293)**
**Huggingface dataset_id:**
**UBC-NLP/palmx_2025_subtask1_culture**
**Input:** ( الإجابة الصحيحة باختيار الحرف الذي يمثل الإجابة الصحيحة B أو C أو D).\n أي\n أجب عن السؤال التالي من اللغات التالية تُستخدم بشكل واسع في قطر بجانب الفرنسية"، n\الإسبانية\n\الإنجليزية\n\الصينية\n\العربية؟
**Output:** B
**Input:** ( ما الإجابة الصحيحة B أو C أو D).\n أي\n أجب عن السؤال التالي باختيار الحرف الذي يمثل الإجابة الصحيحة العامل الأساسي الذي يميز استهلاك قطاع الزراعة للمياه في الأردن مقارنة بالمعدل العالمي للزراعة ارتفاع معدلات الأمطار n\اعتماد القطاع الزراعي على موارد المياه العذبة المحدودة في الأردن\n\تشجيع السياسات الحكومية على توسيع مساحات الزراعة n\السنوية التي توفر مياه إضافية للزراعة\n\استخدام تقنيات الري الحديثة على نطاق واسع في جميع المحاصيل الزراعية"، "n\التقليدية
**Output:** A

**Dataset Demand:**
Search for publicly available or published datasets that meet the following characteristics:\n\n• Language & Format: Modern Standard Arabic. \n• Task Type: Multiple-choice question answering / classification. \n• Domain: Arab general culture (history, traditions, geography, etc.). \n• Record Structure: Each entry contains one culturally focused question plus exactly four answer options labeled A, B, C, D. \n• Expected Annotation: A single capital letter (A, B, C or D) indicating the correct option. \n• Data Source: Fully human-generated questions and answers. \n• Dataset Size: Approximately a few thousand examples (around 2 K samples).

**OpenAI o3 Synthesis Dataset Sample (Accuracy: 0.257)**
**Input:** معاوية بن أبي سفيان\nC. عبد الملك بن مروان\nB. السؤال: من هو مؤسس الدولة الأموية؟\nA. عمر بن عبد العزيز\nD. "يزيد بن معاوية"
**Output:** B
**Input:** سوريا\nD. العراق\nC. الأردن\nB. السؤال: في أي دولة تقع مدينة البتراء الأثرية الشهيرة؟\nA. "مصر"، ""
**Output:** A

**OpenAI gpt-4o Search Dataset Sample (Accuracy: 0.271)**
**Huggingface dataset_id:**
**MBZUAI/ArabicMMLU**
**Input:** You will be given a multiple-choice question in Arabic. Read the context and the question carefully, then select the correct option. Reply with ONLY the letter of the correct choice (A, B, C, D, or E).\n كان هذا اليوم حقًا يومًا سعيدًا، فأنا، والدي\n ووالدتي وأخوتي وأخوات كلنا ذهبنا إلى حديقة الأزهر بالسيارة، فوالدي طول الأسبوع يعمل مهندسًا في الشركة، أما أمي فتعمل في المستشفى، فهي طبيبة، وأنا وإخوتي وأخواتي في المدرسة أو الجامعة. جهزَّت أمي لنا طعامًا شهيًّا، واشترى لنا أبي المثلجات والمقرمشات، وأختي الكبيرة أعدتْ لنا الحلوى والكيك، أما أنا فجهزتُ مع إخوتي العصائر، وغسلنا الفاكهة، ووضعناها في علبة بعد تجفيفها، وجهزت أختي الصغيرة أدوات المائدة والأطباق، وأخي الصغير أحضر معه الكرة والحبل والدراجة والالعاب الورقية. ركبتُ وأسرتي السيارة في الصباح الباكر، وقاد أبي السيارة إلى الحديقة، وعندما وصلنا نزلنا منها وساعدنا أبي وأمي في إعداد الطاولة، ثم ذهبنا نلعب، وجلس أبي مع أمي يتحدثان قليلاً، وضحكنا كثيراً، وأكلنا، وفي المساء قبل اقرأ الفقرة التالية ثم اختر البديل والعشاء عندنا إلى البيت وصلينا العشاء، ثم نمنا، فكان هذا حقا يوم سعيد.\n خرج في هذا اليوم المناسب لـ [فراغ] الذي يكمل الجملة بشكل صحيح\nNone\nالأسرة\n\الأخوات\n\الأم\n\الأب[فراغ] \.
**Output:** D
**Input:** You will be given a multiple-choice question in Arabic. Read the context and the question carefully, then select the correct option. Reply with ONLY the letter of the correct choice (A, B, C, D, or E).\n كان هذا اليوم حقًا يومًا سعيدًا، فأنا، والدي\n ووالدتي وأخوتي وأخوات كلنا ذهبنا إلى حديقة الأز هر بالسيارة، فوالدي طول الأسبوع يعمل مهندسًا في الشركة، أما أمي فتعمل في المستشفى، فهي طبيبة، وأنا وإخوتي وأخواتي في المدرسة أو الجامعة. جهزَّت أمي لنا طعامًا شهيًّا، واشترى لنا أبي المثلجات والمقرمشات، وأختي الكبيرة أعدتْ لنا الحلوى والكيك، أما أنا فجهزتُ مع إخوتي العصائر، وغسلنا الفاكهة، ووضعناها في علبة بعد تجفيفها، وجهزت أختي الصغيرة أدوات المائدة والأطباق، وأخي الصغير أحضر معه الكرة والحبل والدراجة والالعاب الورقية. ركبتُ وأسرتي السيارة في الصباح الباكر، وقاد أبي السيارة إلى الحديقة، وعندما وصلنا نزلنا منها وساعدنا أبي وأمي في إعداد الطاولة، ثم ذهبنا نلعب، وجلس أبي مع أمي يتحدثان قليلاً، وضحكنا كثيراً، وأكلنا، وفي المساء قبل اقرأ الفقرة التالية ثم اختر البديل والعشاء وصلينا العشاء، ثم نمنا، فكان هذا حقا يوم سعيد.\n ذَهَبَ الجميع إلى الحديقة المناسب لـ [فراغ] الذي يكمل الجملة بشكل صحيح\nNone\nالقطار\n\الحافلة\n\السيارة\n\الدراجة[فراغ]ب\.
**Output:** B

Figure 6: **Case study comparing data construction for reasoning-based versus knowledge-based tasks.** The left example (a legal reasoning task) shows that a synthesis agent generates a structured, high-quality output. The right example (an Arabic cultural classification task) demonstrates that a search agent successfully finds a highly relevant existing dataset with broad factual coverage.

format demonstrates superior alignment with the reference set's characteristics, exhibiting greater comprehensiveness and analytical depth. This core advantage stems from its multi-round process, which constructs a more refined analytical perspective and uncovers high-quality datasets that are otherwise difficult to find. This enhances not only the breadth of information but, more importantly, the logical coherence of the resulting dataset, leading to clear performance gains in the final fine-tuning evaluation.

*Deep Research Case*

**Reference Set Sample (Accuracy: 0.33)**
dataset_id: vnhsge/biology
**Input:** Câu 81. Trên một NST thường, khoảng cách giữa hai gen A và B là 34cM. Theo lí thuyết, tần số hoán vị giữa hai gen này là\nA. 33%.\t\nB. 17%.\t\nC. 66%.\t\nD. 34%.
**Output:** D

**Dataset Demand :**
Find a dataset for supervised fine-tuning that contains Vietnamese national exam-style questions spanning multiple academic subjects (literature, English, history, geography, civics, STEM) and includes both multiple-choice questions and literary essays. The dataset should feature multi-modal inputs (text and accompanying images) and use an instruction-input-output format where each sample includes a system directive, an input field with exam texts (and potential image references), an output field with a multiple-choice answer (e.g., a letter), and an output_program field providing detailed reasoning. It must support multi-task evaluation (question answering, text generation, reading comprehension, and visual question answering) and be large-scale (from 10K up to 100K samples) so that it benchmarks LLM performance against high-stakes national exam conditions with domain-specific reasoning and long-context handling.

**OpenAI o3 Synthesis Dataset Sample (Accuracy: 0.05)**
**Input:** <<SYSTEM>> Bạn là trợ lý luyện thi THPT Quốc Gia. Trả lời theo yêu cầu.\n### Bài đọc hiểu (Ngữ văn):\nĐọc đoạn trích sau và trả lời câu hỏi.\n \"...Những đêm sương mùa đông lạnh buốt, Mị vẫn nghe tiếng sáo gọi bạn, tiếng chó sủa làng xa vang vọng Tô Hoài).\nCâu hỏi: Tác dụng nghệ thuật nổi bật của chi tiết \"tiếng sáo gọi bạn\" trong đoạn trích là gì?\nA. Khắc họa bối cảnh thiên nhiên vùng cao.\nB. Thể hiện khát vọng sống, khát khao tự do của nhân vật Mị.\nC. Tái hiện không khí lễ hội tết cổ truyền.\nD. Tô đậm hoàn cảnh nghèo khổ, lạc hậu của người Mông.\n"
**Output:** Đáp án: B\n\nLời giải chi tiết: \"Tiếng sáo gọi bạn\" là một motif âm thanh quen thuộc trong văn học Tây Bắc, gợi nhịp sống rộn ràng của tuổi trẻ và tình yêu. Chi tiết ấy đột ngột vang lên giữa đêm đông giá lạnh, đánh thức sâu thẳm trong Mị khát vọng sống mãnh liệt bị kìm nén bấy lâu. Do đó, nó không chỉ miêu tả không khí lễ hội hay bối cảnh, mà chủ yếu là nghệ thuật thức tỉnh tâm hồn nhân vật, thể hiện khát vọng sống, khát khao tự do.

**OpenAI gpt-4o Search Dataset Sample (Accuracy: 0.005)**
**Huggingface dataset_id:**
testli/English_exam31.1
**Input:** Please read the following instruction and its associated difficulty level, then produce the expected output accordingly. Follow the guidelines carefully.\nGenerate 9 multiple-choice questions for the Vietnamese National High School Graduation Examination in English, specifically for the reading comprehension advanced (requires deeper analysis and inference from text) section. Each question should:\n - Include four answer options.\n - Clearly identify the correct option, ensuring it is one of the provided options (i.e., it must be 0, 1, 2, or 3).\n - Ensure that the exam_section value matches 6.\n Additional guidelines:\n 1. Ensure questions and passages are unique and challenging.\n 2. Use grammatically correct English and avoid duplication of passage meanings, topics, or questions.\n 3. Exclude any harmful, inappropriate, or culturally insensitive content.\n4. Ensure the Flesch-Kincaid Grade level is 11\n11
**Output:** 'text_section_body': \"My favorite Vietnamese festival is Tết Nguyên Đán, often called Tet, the Lunar New Year. It's a time for family reunions, enjoying delicious feasts, and celebrating new beginnings. Preparations begin weeks in advance; families clean their houses meticulously symbolizing washing away the past year's bad luck. Red envelopes containing lucky money are prepared for children and the elders exchange greetings and warm wishes. The family gathers for elaborate meals, and traditional lion and dragon dancers parade through the streets carrying joyous celebration through its many activities. This multi-day celebration fills me with warmth and hope every year.\", \n'question': ['question_text_0: \\'What is the most vital essence of Tet ?\\',\\noptions_of_question_text_0: [\\'The festival takes one day to celebrate.\\', \\'Homes are lightly cleaned up to represent the season.\\', \\'The festival only involves festive meals.\\', \"It\\'s primarily characterized by gathering family, enjoying feast,and welcoming new beginnings\"],\\ncorrect_option_of_question_text_0: 0,\\n', \"question_text_1: 'What reflects deep aspects of celebrating the Tet festival preparing for its activities ?,\\noptions_of_question_text_1: ['Children enjoy no such celebrations.', 'Preparing takes days but the traditions are few.', 'The money given is not particularly special…

**OpenAI DeepResearch Search Dataset Sample (Accuracy: 0.19)**
**Huggingface dataset_id:**
SEACrowd/m3exam
**Input:** I. ĐỌC HIỂU (3,0 điểm)\nĐọc đoạn trích:\n"chính mẹ đẻ anh hùng và truyền thuyết\ntù túp lều lợp lá lợp tranh\ncắt cuống nhau bằng lưỡi liềm\nbàn chân thô quanh năm bùn lấm\nchưa một lần ướm qua sử sách\ntập con bước vịn vào ca dao tục ngữ\ndù uống nước đâu lòng vẫn nhớ nguồn\nthương từ cái kiến con ong\ntím ruột bầm gan thù bọn ác\ndân tộc tôi khi đứng dậy làm người\nlà đứng theo dáng mẹ\n\"đòn gánh tre chín dạn hai vai\" (1)\nmùa hạ gió Lào quăng quật\nmùa đông sắt se gió bắc\ndân tộc tôi khi đứng dậy làm người\nmồ hôi vã một trời sao trên đất\ntrời sao lặn hóa thành muôn mạch nước\ncháy âm thầm cháy dọc thời gian "\n\n(1) Câu thơ của Nguyễn Du. Trích Những người đi tới biển, Thanh Thảo, NXB Quân đội Nhân dân, 2004, tr. 53-54)\nThực hiện các yêu cầu sau:\nCâu 1. Xác định thể thơ của đoạn trích."
**Output:** Thể thơ của đoạn trích: Thể thơ tự do

Figure 7: In this case, deep research agent demonstrates superior performance over DataResearcher with search and synthesis agents.

*Failed on Corner Case*

**Reference Dataset Sample (Accuracy: 0.917)**
**Huggingface dataset_id:**
**ofir408/MedConceptsQA**
**Input:** Question: What is the description of the medical code 89.43 in ICD9PROC?\nA. Enteral infusion of concentrated nutritional substances\nB. Cardiovascular stress test using bicycle ergometer\nC. Measurement of systemic arterial blood gases\nD. Replacement and removal of therapeutic appliances, option1: Enteral infusion of concentrated nutritional substances, option2: Cardiovascular stress test using bicycle ergometer, option3: Measurement of systemic arterial blood gases, option4: Replacement and removal of therapeutic appliances
**Output:** Cardiovascular stress test using bicycle ergometer

**Dataset Demand :**
Search for a dataset designed for few-shot learning evaluation that uses an instruction fine-tuning format with system-input-output triplets. The dataset should consist of detailed multi-choice questions with contextual prompts and enumerated answer options (e.g., A, B, C, D) along with a system directive such as \"Select the best option.\"

The expected output must be a short, full-sentence answer that exactly selects the correct option based on domain-specific content, including professional and academic tasks like medical coding (e.g., ICD9PROC). The collection should encompass 1K-10K examples focused on assessing factual accuracy and generalization in low-resource settings.

**OpenAI o3 Synthesis Dataset Sample (Accuracy: 0.0)**
**Input:** System: You are a certified medical coding assistant. Select the best option.\n\nContext: A 58-year-old patient underwent a laparoscopic cholecystectomy with intraoperative cholangiography for symptomatic gallstones.\n\nQuestion: Which ICD-9-CM procedure code most accurately describes the primary surgical procedure performed?\nOptions:\nA. 51.22 – Cholecystectomy, not otherwise specified\nB. 51.23 – Laparoscopic cholecystectomy\nC. 51.24 – Open partial cholecystectomy\nD. 51.19 – Other excision of gallbladder\n\nTask: Select the best option.
**Output:** The correct option is B.

**OpenAI gpt-4o Search Dataset Sample (Accuracy: 0.067)**
**Huggingface dataset_id:**
**shuyuej/instruction_tuning_data**
**Input:** You are a medical expert. Please read the following instruction and patient case carefully, then select the best answer among the provided options and explain your reasoning in detail.\nYou're a doctor, kindly address the medical queries according to the patient's account.\nAnalyze the question and answer with the best option.\nA 23-year-old pregnant woman at 22 weeks gestation presents with burning upon urination. She states it started 1 day ago and has been worsening despite drinking more water and taking cranberry extract. She otherwise feels well and is followed by a doctor for her pregnancy. Her temperature is 97.7°F (36.5°C), blood pressure is 122/77 mmHg, pulse is 80/min, respirations are 19/min, and oxygen saturation is 98% on room air. Physical exam is notable for an absence of costovertebral angle tenderness and a gravid uterus. Which of the following is the best treatment for this patient?\n A. Ampicillin\n B. Ceftriaxone\n C. Doxycycline\n D. Nitrofurantoin
**Output:** The correct answer is (D).\nExplanation: The patient's presentation is indicative of urinary tract infection (UTI), which is common in pregnant women due to urinary stasis that occurs from progesterone-induced ureteral relaxation and compression of the bladder by the growing uterus. UTI in pregnancy is considered complicated by the inherently increased risk of pyelonephritis in this population and is treated with 7 days of antibiotics regardless of symptoms. Nitrofurantoin, as in option D, is first line for UTI in pregnancy due to its safety profile. It's important to note, though, that it's not used in late pregnancy (i.e., at or after 38 weeks of gestation) or in patients with a creatinine clearance of <40-60 mL/min due to hemolytic anemia risk in the neonate and lack of drug concentration in renal tissue, respectively.\n\nAmpicillin (option A) and Ceftriaxone (option B) are not typically first-line options for uncomplicated cystitis. Doxycycline (option C) should be avoided in pregnancy due to the risk of teratogenicity.

Figure 8: Performance comparison in a challenging corner case. All evaluated methods show degraded performance in this scenario.

## C.3 LIMITATIONS OF CURRENT METHODS FOR CORNER CASES

Despite their power, existing search and synthesis methods are fundamentally constrained by the data distributions they were trained on. Figure 8 illustrates a "corner case"—a task scenario so niche that constructing a suitable training dataset from existing sources is nearly impossible without directly

plagiarizing the reference set (Shoham & Rappoport, 2024). Consequently, the resulting fine-tuning performance is markedly poor.

This limitation is rooted in the data-dependent nature of current agents. Search methods are limited to what is indexed, and synthesis methods are limited to patterns seen during training. Because real-world data distributions are imbalanced and often underrepresent the niche, "corner case" scenarios, models lack the necessary prior knowledge to perform well. This demonstrates an inherent limitation in agents that rely solely on existing data distributions and calls for the development of more flexible and adaptive solutions.

## D   METADATA

Our benchmark provides comprehensive metadata annotations for each task instance to facilitate systematic evaluation and analysis. Each MetaTriplet contains the following structured metadata components:

- **Introduction**: The task and area description of this task instance.

- **Task**: The classification of task type.

- **Question**: Question Content Type - Describes the primary knowledge domains and content types covered by the questions in the test dataset.

- **Input**: Structured retrieval results and contextual information - Input consists of formatted search results containing metadata fields such as descriptions, display URLs, titles, actual URLs, and ranking information, along with potential tabular data, document snippets, and conversational dialogue history for multi-turn scenarios.

- **Output**: Direct factual answer format - Outputs are concise, definitive answers that directly address the question based on the provided context, formatted as complete statements such as 'The answer is [specific fact]' for factual queries, numerical values for arithmetic problems, and explicit acknowledgment when questions cannot be answered.

- **Example**: dataset instance example.

## E   METADATA EVALUATION OF DATASETRESEARCH-PRO

We present the metadata results of six discovery approaches on DatasetResearch-pro through a radar chart, as shown in Figure 9. The chart clearly demonstrates significant performance differences between the two major method categories, particularly highlighting the comprehensive advantages of the Deep Research method in the Search Group across all dimensions, with especially excellent performance in Task and Output Alignment. The Deep Research approach brings comprehensive benefits, particularly in task and output aspects. Notably, generative methods perform most prominently in the Output and Intro dimensions, reflecting the inherent advantages of synthetic data in output quality and instruction-following accuracy for tasks. Furthermore, the comparison between OpenAI o3 w/ ref and w/o ref validates the positive impact of reference samples on improving data generation quality.

## F   PROMPTS

### F.1   PROMPTS FOR DATASET CURATION

This prompt instructs OpenAI o3 to generate a comprehensive dataset demand description by transforming extracted metadata into generic, discoverable descriptions that omit specific identifiers while preserving all essential characteristics needed for effective dataset retrieval.

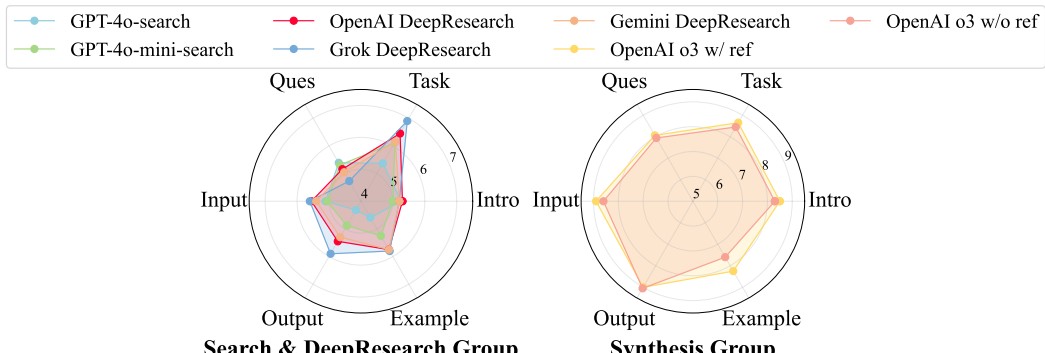

Figure 9: **Radar chart illustrating the average metadata alignment scores for the Search Group, Generation Group, and DeepResearch Group on the `DatasetResearch-pro` subset.** The Generation and DeepResearch groups achieve higher and more balanced scores, indicating superior semantic alignment with the ground-truth dataset requirements.

---

**Prompt for demand description generation**

Imagine you are performing a dataset search task, and the ultimate ground truth (the most ideal dataset found) is the dataset I provided to you. Now, based on your extraction of this dataset's metadata in the previous conversation, generate a prompt for the dataset search task (this prompt will be passed to a powerful deep research agent to complete the task). This prompt needs to include all the metadata from the previous conversation, except for the example.

Previous metadata extraction: {metadata_json}

Please generate a search prompt that a research agent could use to find this exact type of dataset. The prompt should be comprehensive and include all the key characteristics that would help identify this specific dataset type.

IMPORTANT REQUIREMENTS:
1. DO NOT mention the original dataset name or any specific dataset identifiers
2. Include all metadata information EXCEPT the example field
3. For dataset size/samples count, use approximate ranges rather than exact numbers (e.g., "around 10K samples" instead of exact counts)
4. Focus on the task type, domain, input/output characteristics, and source information
5. Make the prompt generic enough that it could find similar datasets, not just the specific one

Output the search prompt directly without any additional formatting or explanation.

---

## F.2 PROMPTS FOR DATA RESEARCHER

This prompt instructs AI to search Hugging Face for publicly accessible datasets and return a specified number of dataset IDs in strict JSON format.

972
973
974
975
976
977
978
979
980
981
982
983
984
985
986
987
988
989
990
991
992
993
994
995
996
997
998
999
1000
1001
1002
1003
1004
1005
1006
1007
1008
1009
1010
1011
1012
1013
1014
1015
1016
1017
1018
1019
1020
1021
1022
1023
1024
1025

**Prompt for search agent**

{agent_query}

IMPORTANT: You must search for publicly accessible datasets from Hugging Face and return exactly {NUM_SEARCH} suitable dataset IDs.

SEARCH STRATEGY:
- First, search for datasets that closely match the query
- If you cannot find enough datasets, gradually relax the search criteria to find more potential datasets
- Include both popular and less popular datasets that might be relevant
- Consider datasets with similar tasks, domains, or data types

OUTPUT FORMAT REQUIREMENTS:
You MUST output your response in EXACTLY this JSON format - do not include any other text or explanations:

{'search_datasets': ['dataset_id_1', 'dataset_id_2', 'dataset_id_3', 'dataset_id_4', 'dataset_id_5', 'dataset_id_6', 'dataset_id_7', 'dataset_id_8', 'dataset_id_9', 'dataset_id_10']}

- Use ONLY the exact format above
- Replace dataset_id_1, dataset_id_2, etc. with actual Hugging Face dataset IDs
- Ensure you provide exactly {NUM_SEARCH} dataset IDs
- Do not include any text before or after the JSON
- The JSON must be valid and parseable

This fine-tuning data extraction prompt guides AI to analyze dataset structure and establish conversion rules for transforming it into fine-tuning format with input-output fields, requiring a mandatory task-specific instruction template.

**Prompt for fine tuning data extraction from huggingface datasets**

You are tasked with analyzing a dataset and determining how to convert it into a fine-tuning format for language models. The fine-tuning data should have exactly two fields: "input" and "output".

{readme_section}
{config_section}
{split_section}{sample_section}

Based on the README content (if available) and the sample data, please analyze and provide the following information:

1. **Selected Config**: Which config/subset was selected from the dataset (output "None" if there's only one config)
2. **Selected Split**: Which split was selected (train/test/validation/etc.)
3. **Conversion Rules**: Describe the rules for converting the dataset samples into input/output format for fine-tuning, including:
- Which columns/fields should be combined for the "input"
- Which columns/fields should be used for the "output"
- **MUST include appropriate instruction text** (e.g., "Please answer the following question:", "Classify the sentiment:", "Translate the text:", etc.)
- The exact format and order for combining the fields
- Any preprocessing needed for the data

**IMPORTANT REQUIREMENTS:**
- The "instruction_template" field MUST contain a clear, specific instruction appropriate for the task
- The instruction should tell the model exactly what to do (e.g., answer questions, classify, translate, summarize, etc.)
- Do NOT use "null" for instruction_template - always provide a meaningful instruction
- The input should clearly guide the model on what output is expected

Please provide your response in the following JSON format:

{
"selected_config": "config_name or None",
"selected_split": "split_name",
"conversion_rules":
{
"input_components": ["list of field names to include in input (use exact field names from sample)"],
"output_components": ["list of field names to include in output (use exact field names from sample)"],
"instruction_template": "REQUIRED: Clear instruction text telling the model what to do",
"input_format": "detailed description of how to format the input",
"output_format": "detailed description of how to format the output",
"example_conversion":
{
"input": "example input with instruction based on the sample",
"output": "example output based on the sample"
}
}
}

Important: Make sure the conversion makes sense for language model fine-tuning and follows common patterns for instruction-following datasets. The instruction_template is MANDATORY and should be task-specific.

This synthesis agent prompt tasks AI with directly synthesizing a specified number of high-quality training samples based on data requirements, aiming to create superior fine-tuning data compared to existing searched datasets.

---

**Prompt for synthesis agent**

You are a specialized expert in fine-tuning data synthesis. You have the following dataset search requirement: {agent_query}

Your task is to directly synthesize {num_data} corresponding examples based on this requirement. The goal is to create synthetic data that, when used for fine-tuning a large language model, will achieve better performance than fine-tuning on existing datasets found through the search.

Here is a reference example for guidance: {example_data}

You MUST output exactly {num_data} samples in JSON list format, where each sample contains only "input" and "output" fields, following this exact format:

{ "input": "...", "output": "..." }, { "input": "...", "output": "..." }, ...

Important requirements:
1. Generate EXACTLY {num_data} examples
2. Each example must have only "input" and "output" fields
3. Follow the task type and domain specified in the search requirement
4. Use the reference example to understand the expected format and style
5. Ensure diversity across your generated examples
6. Focus on creating high-quality data that will improve model performance through fine-tuning

---

### F.3 PROMPTS FOR METADATA EVALUATION

This prompt instructs AI to analyze a HuggingFace dataset's README and sample data to extract comprehensive metadata including task type, domain, input/output descriptions, and source information in structured JSON format.

Prompt for dataset metadata generation

You will see a README file introduction from a HuggingFace dataset and one sample from it. You need to output the following content based on these materials. Please output in JSON format.

README content:
{readme_content}{sample_section}

Please analyze the dataset based on both the README and the sample (prioritize the sample when there are conflicts, as the README might be vague while we've ensured all samples in the dataset are similar to the provided one), and output the following metadata in JSON format:

{
"introduction": "A one-sentence introduction of the dataset content, concise and clear, including key information about task, domain, input and output",
"task": "Directly output one of: multiple-choice, question-answering, summarization, text-classification, text-generation, translation",
"domain": "Directly output the domain the dataset content involves, such as: aerospace, finance, linguistics, politics, sociology, biology, etc.",
"input": "Directly output the dataset's input content, including its language, such as: an English news text for translation, a multiple-choice question in French philosophy domain, etc. (Consider both sample and README, don't be limited by single sample's domain, but also don't be too broad like README)",
"output": "Directly output the dataset's output content, including its language, such as: a number 0 or 1, a letter A/B/C/D, translated Italian text, etc.",
"source": "Directly output the dataset's source: real-world, human-generated, machine-generated, etc.",
"example": "Directly extract the sample provided in the prompt and put it here",
"samples_count": {samples_count}
}

Important: Please strictly follow the above JSON format and provide a comprehensive analysis based on both README and sample data.

This metadata evaluation prompt tasks AI with comparing two dataset metadata objects across multiple dimensions and providing numerical similarity scores (0-10) for each dimension along with an overall average score in JSON format.

---

**Prompt for metadata evaluation**

I need you to compare two dataset metadata and score their matching degree across the following dimensions.

Dimension descriptions:
- introduction: Dataset introduction and overview
- task: Task type (e.g., text-classification, question-answering, etc.)
- domain: Domain field (e.g., finance, politics, biology, etc.)
- input: Description of input content
- output: Description of output content
- source: Data source (e.g., human-generated, machine-generated, etc.)
- example: Sample data
- samples_count: Number of samples

Original dataset metadata:
{original_metadata}

Search dataset metadata:
{search_metadata}

Please score each dimension on a scale of 0-10 for matching degree, where:
- 10 points: Complete match or highly similar
- 0 points: Complete mismatch or opposite
- Output an integer score. If a dimension is missing or meaningless in one or both metadata, mark it as null

Please output the result strictly in the following JSON format:

{
"introduction": score or null,
"task": score or null,
"domain": score or null,
"input": score or null,
"output": score or null,
"source": score or null,
"example": score or null,
"samples_count": score or null,
"average": average score (excluding null values) or null
}

Note:
1. Only output JSON format, do not include any other text
2. Scores must be numbers between 0-10 or null
3. average is the mean of all non-null scores

---

# G   CONFIG FOR SUPERVISED FINE-TUNING

Below is the standard configuration file for supervised fine-tuning using the LlamaFactory framework, based on the Llama-3.1-8B model with bfloat16 precision and full-parameter fine-tuning strategy:

**Config for supervised fine-tuning**

```
bf16: true
cutoff_len: 4096
dataloader_num_workers: 0
dataset: {dataset_id}
ddp_timeout: 180000000
deepspeed: examples/deepspeed/ds_z3_config.json
do_train: true
finetuning_type: full
gradient_accumulation_steps: 2
learning_rate: 1.0e-05
logging_steps: 10
lr_scheduler_type: cosine
max_samples: 1000
model_name_or_path: models/LLama3/Llama-3.1-8B
num_train_epochs: 3.0
output_dir: LLaMA-Factory/results/{task_id}/saves
overwrite_cache: true
overwrite_output_dir: true
per_device_train_batch_size: 1
plot_loss: true
preprocessing_num_workers: 16
report_to: none
resume_from_checkpoint: null
save_only_model: false
save_steps: 1000
stage: sft
template: llama3
trust_remote_code: true
warmup_ratio: 0.1
```

