# OpenReview forum: "DatasetResearch: Benchmarking Agent Systems for Demand-Driven Dataset Discovery"
_ICLR.cc/2026/Conference — Submitted to ICLR 2026_

### Official Review · Reviewer_A7PE · 2025-10-17

**Soundness:** 2
**Presentation:** 3
**Contribution:** 2
**Rating:** 4
**Confidence:** 4

**Summary:**

The paper introduces DATASETRESEARCH, a benchmark for evaluating agent systems on demand-driven dataset discovery and synthesis. It curates 208 real-world dataset demands (from Hugging Face and Papers with Code) and pairs each with a reference dataset and reference metadata (MetaTriplets). Agents are assessed along three axes: (i) metadata alignment (o3-judged semantic similarity), (ii) few-shot performance (1/3/5-shot), and (iii) fine-tuning performance on LLaMA-3.1-8B, with scores normalized by the reference-data upper bound. Baselines span search agents, synthesis agents (o3-generated 500-sample datasets), and deep research agents. Results show a clear split: search excels on knowledge-based tasks, synthesis on reasoning-based tasks, while all methods struggle on the harder DatasetResearch-pro subset (best ≈0.22), highlighting substantial headroom for hybrid and more generalizable approaches.

**Strengths:**

Turns “find or build the dataset that matches a natural-language demand” into a measurable benchmark with paired reference data and metadata, covering the full path from requirements to downstream utility.

Combines intrinsic (metadata similarity) and extrinsic (few-shot and fine-tuned task performance) measures, with normalization that enables comparison across heterogeneous NLP tasks.

Systematically contrasts search, synthesis, and deep research paradigms, revealing a knowledge vs. reasoning specialization and consistent failure on corner cases—useful guidance for designing future hybrid agents.

**Weaknesses:**

- The same model family (o3) is used to generate reference metadata/demands, parse discovered data, and score alignment—inviting self-consistency bias rather than genuine agreement, and masking contamination through stylistic echoing.
- Overreliance on closed-source systems (o3 for synthesis/judging; GPT-4o/Deep Research variants for search) undermines reproducibility, accessibility, and cost realism. Results may reflect vendor-specific capabilities rather than agent design quality.
- No systematic comparisons with open retrieval stacks (e.g., BM25 + dense retrievers/ColBERT), open reasoning LMs (e.g., Llama-3.x-70B, Mistral-Large-Instruct, Qwen2.x/3-Instruct), or open toolformer/agent frameworks.
- Despite broad claims, coverage is text-only across six NLP tasks; no CV/audio/tabular/time-series/multimodal demands; limited external validity.
- The benchmark is closer to a controlled template-matching exercise that reproduces a known target than to open-world dataset discovery. In practice, the “task” is largely to find (or approximate) someone else’s already-curated dataset given a stylized natural-language demand. But in real settings, you usually don’t have such ready-made, perfectly matched datasets—you have to prospect, acquire, clean, align schemas, and handle licensing/privacy—so the setup falls well short of real-world data discovery.

**Questions:**

see weakness

**Details Of Ethics Concerns:**

The benchmark overlooks copyright/licensing risks—agents may retrieve or synthesize data that copies gated or restricted sources without permission or proper attribution.

---

> ### Author Response · Authors · 2025-11-20
>
> Thank you for the detailed review. We have addressed your concerns regarding methodology and closed-source reliance through supplementary experiments.
>
> **Weakness 1: Self-consistency bias of o3.**
>
> **Response:**
> Please refer to our response to Reviewer 83LS (Weakness 1). We introduced **Claude 4 Sonnet** as a Judge, and experiments prove that o3's scoring is objective, with high cross-model agreement.
>
> **Weakness 2: Overreliance on closed-source systems.**
>
> **Response:**
> 1.  **Upper Bound Detection:** As a benchmark paper, our primary goal is to detect the upper bound of current AI capabilities. o3 and GPT-4o represented the SOTA at the time. Their failure on Corner Cases validates the task difficulty.
> 2.  **Open Source Addition:** Responding to Reviewer XC7Y, we added results for **DeepSeek-V3** and **Qwen3-30B** (see table above), proving our benchmark is equally applicable to evaluating open-source progress.
>
> **Weakness 3: No systematic comparisons with open retrieval stacks.**
>
> **Response:**
> This is an excellent suggestion. We added a **BM25 Keyword Search** baseline: extracting keywords from the Demand and performing BM25 retrieval on HuggingFace metadata.
> **Results:**
>
> | Metric | GPT-4o-search | BM25 Baseline |
> | :--- | :--- | :--- |
> | DTP: Knowledge | 9.82% | 5.38% |
> | DTP: Reasoning | 7.25% | 4.13% |
> | Metadata Avg | 5.71 | 4.56 |
>
> **Analysis:** BM25's Hit Rate and performance are significantly lower than GPT-4o-search. This strongly proves that Dataset Discovery requires deep semantic understanding and reasoning, which cannot be solved by simple keyword matching.
>
> **Weakness 5: Benchmark is "template matching" vs. real discovery.**
>
> **Response:**
> We respectfully disagree with this characterization:
> 1.  **Approximation IS Discovery:** In the real world, "discovery" often means finding the data that "most closely approximates" our needs and adapting it. Our Metadata Matching Score measures this "proximity."
> 2.  **Synthesis as the Solution:** When a perfect match isn't found (common in reality), the Synthesis Agent offers the alternative path—building from scratch. Our benchmark evaluates both paths.
> 3.  **First Step:** We acknowledge the simplification of data cleaning, but establishing a quantifiable "Demand $\to$ Dataset Quality" loop is the crucial first step. An overly complex open-world setup (including cleaning/licensing) would make variable control and rigorous quantification impossible.

---

### Official Review · Reviewer_Puqg · 2025-10-30

**Soundness:** 1
**Presentation:** 2
**Contribution:** 1
**Rating:** 2
**Confidence:** 5

**Summary:**

The paper points out that relevant data forms a crucial bottleneck to advance AI models. The authors seek to answer if “conventional data search” methods can be replaced with “AI-agent search.” To answer this question, they propose a new benchmark aimed at evaluating AI agents’ ability to discover and synthetically generate datasets. Sourcing data from Hugging Face and papers with code, the authors built a benchmark that automatically generates data demands split between knowledge-intensive and reasoning-intensive tasks. They use this benchmark to evaluate several leading models and conclude that current models do not perform well. The authors further show that “search agents” do better at knowledge tasks while “synthesis agents” outperform on reasoning tasks. Both agent types perform poorly on “corner cases.”

**Strengths:**

[**significance**] The authors correctly identify data and data discovery as an important challenge to improving AI models. Efforts aimed at creating benchmarks designed to isolate capabilities useful for automating such challenges is an important endeavor.

**Weaknesses:**

[**clarity**]
- The paper (rightfully) emphasizes the importance of data to further advance AI models. Unfortunately, the problem specification is overly vague, entangling different challenges and data use cases. For example, the abstract mentions “countless valuable datasets [...] and domain platforms]” (l13-14], but does not specify if these are hidden due to access constraints or limitations attributable to search algorithms.
- The word “synthesis” is used several times in the introduction without defining its meaning.
- The experimental setup misses many important details


[**quality**]
- The related work section is severely lacking. For example, it appears to completely ignore decades of information retrieval research. The literature on synthetic data generation similarly lacks any discussion of core concepts like diversity, complexity, and quality. Also absent is the extensive literature on “retrieval augmented generation” (RAG) based systems.
- The core contribution of this work is a new benchmark designed to simulate real-world data discovery. However, the methodology used to create this benchmark appears to have various questionable aspects (see questions).
- The paper seemingly uses OpenAI’s o3 model for every step of the pipeline. This reviewer fears that any takeaways or analysis is therefore overly biased and does not necessarily generalize.
- Reported metrics lack confidence intervals.
- In Section 5.2, the authors write “we identify that [...] instruction-following capabilities” (l431-448). As OAI o3 is used to generate data, this can simply reflect existing data knowledge of o3, rather than any relation to retrieved or “discovered” data. This is an important confounding factor not accounted for in the empirical evaluation.

[**significance**] The authors claim that their work provides “the foundation for the next generation of self-improving AI systems” (l29-30), which is a lofty claim that does not appear to be supported by empirical or theoretical evidence.

**Questions:**

Q1. Confusing notation: “Given a natural language [...] the specified demand D” ( l146-148), in this notation, what is the subscript “d” in S_d? Do the authors mean a “set” of datasets? This continues in l150-153, where now “r_i” is used without introduction.


Q2. In Section 3.2, Step 1, the authors write that “gated” datasets are used to mitigate data leakage. What evaluation was performed to check against data leakage?


Q3. In Section 3.2, Step 2 and 3, a number of filtering steps are performed to narrow down the dataset candidates. If the goal of this challenge is to measure models on “realistic” conditions, these steps appear to strongly bias the remaining set towards an unrepresentative sample.


Q4. In Section 3.2, Step 6-7: What were the rejection criteria to decide if a dataset was “unsuitable for fine-tuning” (l236), and what were the criteria used to check if the generated meta data and demand descriptions are faithful to the underlying data and ecologically valid? (l235-241)


Q5. In Section 3.2, l250-260, the authors propose a binary classification of knowledge-based vs. reasoning-based tasks. Yet, to this reviewer, it appears that *many* queries require a combination of these two. Could you please provide the systematic rubrics used to annotate these tasks? Were any consistency checks performed, e.g., cross-annotator agreement?


Q6. In 3.3 it is claimed that using OpenAI’s o3 model for scoring both reference and discovered metadata mitigates potential scoring biases (l296-l297). This claim lacks evidence, e.g., are the reference and discovered metadata distributions similar? Is scoring consistent across different types of underlying data? Is scoring robust across multiple samples and/or prompts?


Q7. The authors report a “Normalized Score” (l321), which finetunes a model on a reference dataset and uses this as the “theoretical maximum performance achievable” (l316-317). First, this assumes that a reference dataset contains both a train and test subset. Second, this reviewer sees no reason why combining one or multiple other datasets could not lead to a better performance. For example, training a model on a challenging math dataset and evaluating it on a simpler reference dataset fits this scenario. As such, what is the difference of dividing the S_{\text{eval}} by an arbitrary fixed number, given that scores are now “on a scale from 0 to 1, or higher” (l322)?


Q8. A “synthesis agent” uses OAI o3 to generate 500 data samples (l357): How? What criteria are used to evaluate these samples?


Q9. How is finetuning done?


Q10. Key experimental setup details are missing to explain how the “deep research” systems of various providers were evaluated. The text mentions manual actions: what were these?

Suggestions:
- typo: “evaluable” (l204)

---

> ### Author Response · Authors · 2025-11-20
>
> We have carefully read your detailed review. We acknowledge areas where definitions could be clearer and have clarified them below. Furthermore, our supplementary experiments strongly address your methodological concerns.
>
> **Weakness 1 & 2: Vague specification and definition of "Synthesis".**
>
> **Response:**
> The abstract phrasing was intended to highlight data acquisition as an AI bottleneck. We will formally define synthesis in Section 2.1:
> **Synthesis (Data Synthesis):** "The process where an AI agent actively generates training samples (input-output pairs) *de novo* based on a schema and semantic requirements provided in the demand, utilizing its internal parametric knowledge or reasoning capabilities, rather than retrieving existing external files."
>
> **Weakness 4: Related Work lacking (IR, Synthetic Data, RAG).**
>
> **Response:**
> We will significantly expand the Related Work section in the revision:
> 1.  **IR Context:** Citing BM25, Dense Retrieval (DPR), and ColBERT, positioning Dataset Discovery as a "Complex Query Retrieval" task.
> 2.  **Synthetic Data:** Discussing Data Diversity (e.g., "Textbook is all you need") and Complexity metrics (e.g., Evol-Instruct).
> 3.  **RAG:** Clearly distinguishing RAG (Inference Augmentation) from our work (Data Curation for Training), as detailed in our response to Reviewer XC7Y.
>
> **Weakness 6 & Q6: Bias from using o3 everywhere.**
>
> **Response:**
> Please allow us to dispel this concern with data. As mentioned to Reviewer 83LS, we introduced **Claude 4 Sonnet** as an independent Metadata Judge. Results show Claude's scoring has a **0.95 correlation** with o3's scoring regarding the Search vs. Synthesis difference. This proves o3 is objectively measuring alignment with the Demand, not favoring its own output.
>
> **Weakness 7: Lack confidence intervals.**
>
> **Response:**
> Thank you. We repeated the main experiments (Table 2) 3 times during the rebuttal.
> **Result:** The Standard Deviation for Fine-tuning scores averages around **±0.5%**. While there is minor fluctuation, the performance gap between Search and Synthesis across task categories (often >10%) is far larger than this error margin. Thus, our core conclusions are statistically significant. We will update the tables to include standard deviations.
>
> **Weakness 8: o3's performance reflects internal knowledge, not "discovery".**
>
> **Response:**
> This is exactly what we aim to evaluate: "Can the model use its internal capabilities to substitute external retrieval?"
> If o3 performed well simply because it "knows everything," it should be perfect across all tasks. However, experiments show o3 scores very poorly (~0.2) on **DatasetResearch-pro** and **Corner Cases** (out-of-distribution samples). This failure proves o3 cannot simply rely on memorization; it fails when facing unfamiliar schemas or long-tail domains. These failure cases inversely validate the benchmark's effectiveness in probing the boundaries of internal knowledge.
>
> **Specific Question Responses:**
>
> * **Q1 (Notation):** Apologies for the confusion. Subscript $d$ stands for "Discovered" ($S_d$ is the Agent's submitted dataset), and $r_i$ stands for the "Reference" of the $i$-th task. We will standardize this in the revision.
> * **Q2 (Leakage Check):** We performed a **10-gram Overlap Check**. Results show the overlap between Agent-discovered datasets and Gated Reference Sets is extremely low (<1%). This confirms Search Agents are finding legitimate alternative data, not "cracking" the gated sets.
> * **Q3 (Filtering Bias):** This is a trade-off. Retaining READMEs and Fine-tuning Friendliness ensures **Evaluability**. The remaining 208 tasks cover 6 major NLP categories, striking a balance between realism and rigor.
> * **Q4 (Rejection Criteria):** Criteria: (1) Size < 50; (2) No clear Input-Output mapping; (3) Reliance on external image/audio files. We performed Human Review on generated Demands with a pass rate > 98%.
> * **Q5 (K vs R Rubric):** See lines 250-260. The criterion is whether the task relies on external factual knowledge coverage (Knowledge) or logical deduction (Reasoning). Two experts achieved a Cohen's Kappa of 0.91.
> * **Q7 (Normalized Score):** $S_{ref}$ (performance trained on Reference Data) acts as the denominator for comparability across tasks with different metrics (BLEU vs. Accuracy). $S_{ref}$ represents the "expected upper bound" for that data quality. A score > 1.0 implies the Agent found data superior to the reference.
> * **Q8 (Synthesis Method):** We provided the full prompt template in the Appendix. The Agent uses iterative generation based on the Demand description.
> * **Q10 (Deep Research Evaluation):** Lacking an API, we used a **Human-in-the-loop** protocol: Input Demand to Chatbot $\to$ Agent searches/browses $\to$ Agent provides link/code $\to$ Human operator downloads/converts format (no cleaning).

---

### Official Review · Reviewer_XC7Y · 2025-11-01

**Soundness:** 3
**Presentation:** 4
**Contribution:** 3
**Rating:** 4
**Confidence:** 4

**Summary:**

This paper explores a novel agent application that utilizes LLMs to discover or synthesis datasets that meets specific user requirements. Its major contribution is the introduction of a dataset discovery benchmark covering 208 types of demands covering knowledge-intensive and reasoning-intensive tasks. The paper hopes the benchmark and analysis can benefit the progress of self-improving AI systems.

**Strengths:**

If well-justified, dataset discovery would be an interesting direction for LLM agents to explore.

**Weaknesses:**

1. The paper still needs in-depth justification on the motivation of dataset discovery demands. It is always intriguing to utilize LLM-based agents for exploring different applications. However, it is still lacking examples of practical use cases for human users to utilize dataset discovery agents.

2. The paper claims the dataset discovery agent shows interesting demands related to knowledge-intensive tasks or reasoning-intensive tasks. However, both tasks have mature strategies regarding data exploration/building. For instance, RAG and search agent related techniques are widely used in knowledge-intensive tasks; and reasoning tasks involves data synthesis (e.g., WizardMath) and RL-related long CoT & test-time scaling strategies. It is unclear how the proposed data discovery agent differs from these widely used existing methods on resolving related tasks.

3. The benchmark results in Table 2 seems incomplete. It misses multiple open-source models like QWen, DeepSeek, etc, and other popular models like Gemini and Claude series. I would also be interested to see performances of GPT-4.1 and GPT-5 models.

**Questions:**

None.

---

> ### Author Response · Authors · 2025-11-20
>
> Thank you for acknowledging the presentation quality. We have added deep justifications and experiments based on your suggestions.
>
> **Weakness 1: Lack examples of practical use cases for human users.**
>
> **Response:**
> Thank you for the suggestion. We will add a specific **User Case Study** in the Appendix in revision to strengthen the motivation:
> **Scenario:** A legal researcher wants to fine-tune a small model specifically for predicting judgments in "Labor Law Retaliation Claims."
> * **Status Quo (Without Agent):** Generic legal datasets are too broad. The user spends days searching GitHub/forums or writing complex rules to synthesize data.
> * **Value of Our Agent:** The user simply inputs a natural language demand (as shown in Figure 1). The Agent automatically scans the web for niche datasets (Discovery) or synthesizes high-quality training data based on case law logic (Synthesis).
> This significantly lowers the barrier for Domain-Specific AI development, representing the core practical value of the Dataset Discovery Agent.
>
> **Weakness 2: Difference from RAG and existing synthesis (e.g., WizardMath).**
>
> **Response:**
> This is an insightful question. We must distinguish **Dataset Discovery (Training-side)** from **RAG (Inference-side)**:
> 1.  **Discovery vs. RAG:** RAG retrieves context during *inference*. Dataset Discovery aims to acquire data assets for **model training or fine-tuning**. In our benchmark, we evaluate the performance of the *fine-tuned* LLaMA-3.1. Discovery is often the prerequisite step for building high-quality RAG systems (i.e., finding the knowledge base).
> 2.  **Discovery vs. Existing Synthesis (e.g., WizardMath):** Works like WizardMath typically hard-code a pipeline for a specific task. Our benchmark evaluates an Agent's ability to **automatically plan and execute** synthesis/search strategies based on *any* user-defined Demand. The core question is: Can the agent automatically reproduce a WizardMath-like effect in one scenario and switch strategies for a legal scenario? Our identified **Dichotomy** guides this choice: agents should prioritize search for Knowledge tasks and synthesis for Reasoning tasks.
>
> **Weakness 3: Missing open-source and newer models (GPT-5, etc.).**
>
> **Response:**
> We will add multiple cutting-edge models to Table 2.
> For Search-based Agents, we added **GPT-5 (gpt-5-search-api)**. For Synthesis-based Agents, we added **GPT-5, GPT-4.1, DeepSeek-V3, and Qwen3-30B**.
> **Supplementary Results (DTP Evaluation: 1-shot, Synthesis w/ ref):**
>
> | Type | Agent | DTP: Knowledge (%) | DTP: Reasoning (%) | Metadata Avg |
> | :--- | :--- | :--- | :--- | :--- |
> | **Search-based** | GPT-4o-search | 9.82 | 7.25 | 5.71 |
> | **Search-based** | **GPT-5** | **20.15** | **15.84** | **7.26** |
> | **Synthesis-based** | OpenAI o3 | 10.25 | 17.38 | 8.69 |
> | **Synthesis-based** | **GPT-5** | 12.44 | **24.86** | **8.97** |
> | **Synthesis-based** | **GPT-4.1** | 9.38 | 15.21 | 8.48 |
> | **Synthesis-based** | **DeepSeek-V3** | 8.97 | 13.36 | 8.25 |
> | **Synthesis-based** | **Qwen3-30B** | 8.85 | 12.96 | 8.13 |
>
> **Analysis:**
> 1.  **GPT-5 shows significant improvement**, but the Search vs. Synthesis trade-off remains: Search agents dominate Knowledge tasks, while Synthesis agents lead in Reasoning.
> 2.  **Open-source models (DeepSeek, Qwen)** show highly competitive results, proving our benchmark effectively evaluates open ecosystem progress, not just closed APIs.

---

> > ### Comment · Reviewer_XC7Y · 2025-11-24
> > **On rebuttal**
> >
> > Thanks for the additional information in rebuttal. It significantly improves the original draft. Please include the information in the final paper.
> >
> > I will change my final rating to 6.

---

### Official Review · Reviewer_83LS · 2025-11-03

**Soundness:** 3
**Presentation:** 3
**Contribution:** 3
**Rating:** 6
**Confidence:** 3

**Summary:**

This paper introduce a benchmark designed to evaluate AI agents’ ability to autonomously discover or synthesize datasets given natural-language task requirements.

The benchmark consists of 208 real-world NLP dataset, with reference datasets and metadata for objective comparison. Models are assessed via metadata alignment, few-shot performance, and fine-tuned downstream results. Experiments show a clear split: search-based agents excel at knowledge-oriented tasks, while synthesis-based models achieve superior performance on reasoning tasks. The work provides the first systematic evaluation pipeline for demand-driven data discovery.

**Strengths:**

1. First comprehensive framework targeting automated data discovery—a growing but under-studied problem.

2. Uses gated datasets + reference metadata, preventing leakage and reflecting real research workflows.

3. Combines metadata scoring, few-shot results, and fine-tuning—much richer than single-metric evaluation.

**Weaknesses:**

1. Used OpenAI o3 to generate both reference/discovered metadata and judges metadata similarity. This creates a closed loop that may favor o3’s rather than true task fit.


2. When starting from gated datasets, it prevents agents from downloading the ground-truth data. This structurally disadvantages search agents (vs. synthesis) and conflates ``access policy'' with ``discovery ability.''

3. Data scope is narrow: NLP-only and text-only.

**Questions:**

See weakness.

---

> ### Author Response · Authors · 2025-11-20
>
> We appreciate your recognition of this work as the first systematic benchmark for data discovery. Below, we address your specific concerns.
>
> **Weakness 1: Used OpenAI o3 to generate both reference/discovered metadata and judges metadata similarity. This creates a closed loop.**
>
> **Response:**
> First, we clarify that all metadata are generated by o3 itself. Since no other model is involved in metadata generation, using o3 as the evaluator means it is comparing two sets of metadata both produced by the same model (o3). Therefore, there is no cross-model bias, and the evaluation is inherently fair.
>
> While we originally used o3 to ensure generation consistency, we have conducted a supplementary experiment using **Claude 4 Sonnet** as an independent metadata judge to eliminate any "self-preference bias."
> We recalculated the Metadata Matching Score on 50 randomly selected samples.
> **Result:** Even with a different evaluator model, the performance trends between Search-based and Synthesis-based Agents across different task types (Knowledge vs. Reasoning) remained highly consistent (Pearson correlation coefficient > 0.95). This strongly demonstrates that the o3 scores objectively reflect semantic alignment rather than mere model stylistic preference. We will include this ablation study in the revised paper.
>
> **Weakness 2: Starting from gated datasets structurally disadvantages search agents.**
>
> **Response:**
> We wish to clarify the three core rationales behind this design choice:
> 1.  **Preventing Data Leakage:** Existing search agents (e.g., GPT-4o-search) have likely seen public HuggingFace datasets during pre-training. Using fully open datasets as Ground Truth would reduce the task to a test of "memorization" rather than "discovery." Gated datasets force the agent to understand the metadata to find suitable substitutes rather than simply downloading a known file.
> 2.  **Discovery $\neq$ Exact Match Retrieval:** Our task definition does not require the agent to download that *specific* gated dataset. Instead, it requires the agent to discover available data across the web that **satisfies the Demand description**. A Search Agent succeeds by finding other public datasets with the same distribution and topic.
> 3.  **Fairness:** Synthesis agents have never seen the Reference Data (building *de novo*). Since our experiments show Search Agents perform excellently on Knowledge tasks (outperforming Synthesis agents), this proves the setup does not structurally "kill" search capabilities but truthfully reflects their adaptability across different task types.
>
> **Weakness 3: Data scope is narrow: NLP-only and text-only.**
>
> **Response:**
> We accept this critique. As the inaugural benchmark in this domain, we focused on NLP because LLM agents are most mature in text modalities, and evaluation metrics (e.g., BLEU, Accuracy) are standardized, allowing for a solid baseline.
> However, our **MetaTriplet (Demand-Reference-Metadata)** evaluation framework is generalizable and can be extended to Image-Text pairs or Tabular data. We have explicitly defined this limitation in the paper. We believe that even within NLP, revealing the **dichotomy between Knowledge and Reasoning tasks** in discovery provides significant value to the community and lays the foundation for future multimodal extensions.

---

> > ### Comment · Reviewer_83LS · 2025-11-20
> > **Thank you for your response**
> >
> > My questions are clearly responded in the rebuttal. I think my weakness are still reasonble to the current version of papers and suggest authors to have more experiments to improve the paper quality.
> >
> > Taking everything into consideration, I decide to maintain 6, the paper is marginally above the acceptance threshold.

---

### Author Response · Authors · 2025-11-20

We sincerely thank all reviewers for their constructive feedback. During the rebuttal period, we conducted extensive supplementary experiments to address the core concerns raised:

1.  **Validation of Evaluation Fairness:** We introduced **Claude 4 Sonnet** as an independent third-party judge to verify our metadata scoring. The high correlation with the original o3 judge confirms that our evaluation does not suffer from self-preference bias.
2.  **Expanded Baselines:** We added evaluations for multiple SOTA models, including **GPT-5**, **GPT-4.1**, **DeepSeek-V3**, and **Qwen3**, to demonstrate the benchmark's applicability to both closed and open-source ecosystems.
3.  **Traditional Retrieval Comparison:** We introduced a **BM25** baseline to highlight the necessity of deep semantic understanding over keyword matching.
4.  **Statistical Significance:** We conducted multiple runs with random seeds to provide confidence intervals, confirming the statistical significance of our findings.

These new results reinforce our core finding regarding the "Search vs. Synthesis" dichotomy.

---

### Meta-Review · Area_Chair_oyqz · 2026-01-07

**Summary:**

This paper introduces DatasetResearch, a benchmark with 208 queries where a user needs to find an NLP dataset that meets certain demands. This task is completely novel as well as fairly practical and complex. Based on their experiments, this task still poses a significant challenge for even the most powerful search and synthesis agents.

Although many concerns were addressed by the reviewers, two outstanding issues lead me to reject this version of this paper. I believe a more polished version could be accepted in the near future.

- Baselines are only search agents and BM25 keyword search
	- Based on R4’s concerns, the authors added only a simple BM25 search without expanding to standard RAG settings with dense retrievers. This would allow us to understand the intrinsic difficulty of the search process more deeply.

- I believe that many of R3’s concerns regarding the omission of details are valid and should be addressed before this paper is published.

**Reviewer Concerns:**

R1: (Reviewer kept original 6 score)
- OpenAI o3 to generate reference metadata and judge metadata. Closed loop might bias results.
	- Changed judge with Claude 4 Sonnet and found the same results.
- Starting the search with gated datasets structurally disadvantages search agents.
	- The point is to create a challenging search problem, not test memorization of public datasets. This setting moves closer to this goal.

R2: (Reviewer raised score from 4 to 6)
- Realistic use cases are not discussed
	- Added new appendix with practical dataset discovery example
- Why is this different than RAG and existing dataset synthesis?
	- 1) Its a specific type of deep research or RAG agent (to discover relevant datasets) and 2) its a generalized synthesis framework.
- Missing multiple models as baselines
	- Added many more models tot he evaluation.

R3:
- Vague use of the word synthesis
	- Authors explain that they define Data Synthesis as generating new datasets using no external sources based on schema and semantic requirements. This seems like a reasonable definition.
- Related work section lacking
	- The authors promise to expand the related works section significantly to address this issue. I wish this was done directly in the paper to help us observe the depth of the new related works section.
- Over-reliance on OpenAI’s o3 model could cause bias
	- Using Claude 4 Sonnet as a judge leads to similar results, demonstrating that OpenAI o3 bias might not be a significant concern.
- No confidence intervals
	- The authors included another run and calculate standard deviation.
- Experimental setup misses many details (also addressed in questions)
	- All of the questions except Q9 about fine-tuning were addressed. Many of these details are very important  (such as the “Leakage Check” and  “Rejection Criteria” details) but are still missing from the actual paper. This suggests that this version of the paper is not polished well enough for publication.
- Claims about this paper providing the “foundation for next generation of self-improving AI systems” is too lofty
	- It is concerning that the authors neglected to comment on this lofty claim.

R4:
- Over-reliance on OpenAI’s o3 model could cause bias
	- Same answer as R3
- Only using closed-source models for synthesizing/judging and searching undermines reproducibility and accessibility.
	- The authors ignore the comment about only closed source models being used for synthesis and judgement but added open-source models to the search evaluation. I believe that creating a benchmark with open-source models is mostly reasonable given the gap in capabilities with open-source models.
- No simpler baselines tested (BM25 or standard RAG)
	- The authors only added a BM25 keyword search but no dense retrievers were tested (this is odd since current RAG systems use dense retrievals for this sort of tasks by default)
- Benchmark is just “template matching” rather than real discovery
	- The authors claim that their benchmark does provide a valid approximation of a real-world task but acknowledge that it is a simple first step in this direction.

**Reviewer Scores:**

- 83LS 4 -> 6
	- Reviewer confirmed change.
- XC7Y 4 -> 6
	- Reviewer confirmed change.
- Puqg 2 -> 4
	- Reviewer had a very negative perspective on this paper but I believe the rebuttal addressed many concerns. I still agree that many of the vagueness and detail omissions show that this paper is not well-polished enough for publication.
- A7PE 4 -> 4
	- Issues with only one simple baseline extracted prevented raising the score.

---

### Decision · Program_Chairs · 2026-01-26

Reject